# The Effects of Cancer Beliefs and Sociodemographic Factors on Colorectal Cancer Screening Behaviours in Newfoundland and Labrador

**DOI:** 10.3390/healthcare10122574

**Published:** 2022-12-19

**Authors:** Yujia Kong, Lance Garrett Shaver, Fuyan Shi, Huaxia Mu, Weixiao Bu, Holly Etchegary, Kris Aubrey-Bassler, Shabnam Asghari, Yanqing Yi, Peizhong Peter Wang

**Affiliations:** 1School of Public Health, Weifang Medical University, Weifang 261053, China; 2Division of Community Health and Humanities, Faculty of Medicine, Memorial University of Newfoundland, St. John’s, NL 1AB 3V6, Canada; 3Faculty of Medicine, University of British Columbia, Vancouver, BC V6T 1Z3, Canada; 4Primary Healthcare Research Unit, Discipline of Family Medicine, Memorial University of Newfoundland, St John’s, NL A1B 3V6, Canada

**Keywords:** colorectal cancer, cancer beliefs, cancer screening

## Abstract

Objectives: This study investigated the beliefs about cancer treatment, outcomes, and screening among adults aged 50–74 in Newfoundland and Labrador and whether these beliefs or sociodemographic factors were associated with differences in colorectal cancer (CRC) screening behaviours. Methods: This analysis uses data collected from an online survey of adults on cancer awareness and prevention in NL. Chi-square tests were used to assess differences in distributions of beliefs based on CRC screening behaviour. Logistic regression was used to identify sociodemographic factors independently associated with CRC screening behaviour. Results: A total of 724 participants were included in the analysis, 57.4% of which had ever had CRC screening. Most held positive beliefs about cancer outcomes and treatment. Only beliefs about screening affected CRC screening behaviour. People who never had CRC screening were more likely to believe their worries about what might be found would prevent them from screening (χ^2^ = 9.380, *p* = 0.009); screening is only necessary if they have symptoms (χ^2^ = 15.680, *p* < 0.001); and screening has a high risk of leading to unnecessary surgery (χ^2^ = 6.824, *p* = 0.032). Regression identified that men had higher likelihood of having had CRC screening than women in our study (OR = 1.689, 95%CI = 1.135–2.515), as did all age groups compared to ages 50–54. No associations were found with the other sociodemographic factors studied. Conclusion: Beliefs about cancer screening appear to play some role in CRC screening behaviour, but the absolute effect was small. The relatively few sociodemographic associations with screening behaviour suggest that NL’s CRC screening program is equitably reaching people from different socioeconomic backgrounds.

## 1. Introduction

Colorectal cancer (CRC) is one of the most common cancers worldwide, and is a leading cause of cancer death; second only to lung cancer in Canada [1,2,3]. The five-year survival rate of colorectal cancer in Canada has shown slow and steady improvement, recently estimated to be 66.8% [2]. In the province of Newfoundland and Labrador, the incidence rate is considerably higher than the rest of Canada, and this is considered to be related to, in part, poor diet and related health behaviours, as well as genetic factors [4,5].

Fortunately, there are robust screening tests available to detect pre-malignant adenomas early, effectively reducing CRC incidence [6]. Among adults aged 50–74, screening with sigmoidoscopy is recommended every ten years, while fecal immunochemical testing (FIT) or fecal occult blood testing (FOBT) is recommended every two years [6]. In NL, the provincial colon cancer screening program, which began as a regional initiative in 2012 and expanded to the rest of the province over the following three years, uses biannual FIT tests for individuals aged 50–74 who are at a normal risk of CRC. Individuals can request a kit through their family physician, online, or by phone. Kits are mailed to the patient who can then complete the test at home and mail it to the lab for testing. There was a considerable push for participation, including with local media, for this program as it was implemented. In the first three years since implementation in 2012, over 10,300 kits were mailed to residents and 76% were returned [7].

There are known associations between intent to screen and factors such as age, knowledge, attitudes, and worries related to cancer and screening tests [8]. Based on this, the authors recommended that interventions aimed at positively influencing knowledge, attitudes, and worries may help improve uptake of CRC screening [8]. Furthermore, the literature also suggests that feeling healthy and fearing test outcomes have been associated with CRC screening avoidance, while factors such as knowing someone with cancer can increase uptake [9]. There are also known socioeconomic gradients in cancer screening participation, and that this relationship is mediated in part by psychosocial factors such as cancer fatalism, life stressors, lack of educational opportunities, limited resources, and low perceived benefits to screening [10].

No studies to date have attempted to assess awareness and beliefs about cancer treatment, outcomes, and screening among the population of Newfoundland and Labrador. The uptake of screening has been far less than desirable in Canada, and it is particularly poor in the region of Newfoundland and Labrador [11]. Across 43 health regions in Canada, never-use rates of FOBT ranged from 53.3 to 89.2%, and never-use of endoscopy ranged from 81.1 to 94.3%, whereas the never-use rates in NL health authorities ranged from 83.6 to 89.1% for FOBT and 84.6 to 94.3% for endoscopy [11], though this was before introduction of the new FIT screening kits. Therefore, understanding reasons for this poor uptake and identifying factors associated with adoption of CRC screening will have value in population health promotion. While beliefs have been found to be associated with screening intent elsewhere, we believed that it was important to survey residents of NL in order to quantify their beliefs and the effects they have on participation in CRC screening, with the hope that this could also help to explain the low rates of CRC screening participation in this province. Furthermore, with the ease at which FOBT screening kits can be requested and completed thanks to NL’s new CRC screening program, we believed it was important to survey residents to identify any associated barriers that may still exist.

The aim of the study was to assess attitudes towards cancer treatment and outcomes, and towards cancer screening, in NL adults aged 50 to 74, and to determine factors affecting CRC screening behaviour. To achieve these aims, three specific objectives were proposed: (1) Describe and analyse associations between participants’ beliefs about cancer (treatment and outcomes) and CRC screening behaviour (having ever had versus having never had CRC screening); (2) Describe and analyse associations between participants’ beliefs about cancer screening and CRC screening behaviour; (3) Analyse independent associations between sociodemographic factors and CRC screening behaviour. We hypothesised that people with more positive beliefs, and fewer negative beliefs, towards cancer and cancer screening would be more likely to participate in CRC screening. We further hypothesized that participants with a family history of cancer, older participants, and participants with higher levels of education, would be more likely to participate in CRC screening. This study was conducted as part of a larger study on the awareness and prevention of cancer among adults aged 35–74 in Newfoundland and Labrador.

## 2. Methods

### 2.1. Design

NL residents aged 35 to 74 years were recruited through Facebook advertising for a web-based survey about cancer awareness and prevention during April and May 2018 (Appendix A). Purposive quota sampling was employed using targeted advertising to improve representativeness of the sample. During the recruitment process, we regularly evaluated the distribution of respondent demographics to find underrepresented samples; targeted advertisements based on geography, age, gender, and education were then employed to improve representation. Not all quotas were met by the time the survey period ended, so some populations were still underrepresented (for details on design and an assessment of sample representativeness, see [12]). The sample was over-representative of women, those with post-secondary education, and those who had an up-to-date flu shot. It was representative of rural-urban geography, those with a regular healthcare provider, and those who have had a colonoscopy [12]. The sample was practically representative of age distribution, household income, smoking status, and BMI [12]. Participation was anonymous, and informed consent was required before participants could start the survey. This study was approved by the Health Research Ethics Authority at Memorial University. We restricted our analyses in this paper to the participants who were between the ages of 50 and 74, as these are the ages included in provincial CRC screening programs.

### 2.2. Data Collection

#### 2.2.1. Participant Information

The questionnaire had 10 questions measuring sociodemographic characteristics such as gender, age group, education, geography, income, and whether they were living with a partner.

#### 2.2.2. Assessing Health and Health Care

These forms were developed to assess participants’ self-rated health, number of chronic illnesses, self-rated life stress, cancer history of among self and others. To assess CRC screening behaviour, participants were asked whether they had ever had (1) an FIT or FOBT test, or (2) a sigmoidoscopy as a screening test for colorectal cancer. Scores were recorded as 0 = Never had screening, and 1 = Ever had screening, for each test. If an individual had any one of the two screening tests, or both, they were classified as having had CRC screening. Individuals who responded “Never had one” to both of the questions, or who responded “Never had one” to one of the questions and left the other blank, were classified as “never had any CRC screening”. Those who did not respond to either screening question were coded as “system missing.” Colonoscopies are not recommended as a screening modality by the Canadian Task Force on Preventive Health Care; therefore, we did not consider having had a colonoscopy as CRC screening.

#### 2.2.3. Assessing Beliefs about Cancer Treatment, Outcomes, and Screening

Participants’ beliefs about cancer treatment and outcomes, as well as beliefs about cancer screening, were assessed. Questions were adapted from a variety of sources, including the Awareness and Beliefs about Cancer (ABC) instrument [13,14,15]. Six questions were from the ABC instrument assessed beliefs about cancer treatment and outcomes, three of which were positive beliefs and three of which were negative beliefs, on a four-point Likert-type scale. Positive beliefs were scored from 1 to 2 (1 = disagree or strongly disagree; 2 = agree or strongly agree); negative beliefs were reverse-scored (2 = disagree or strongly disagree; 1 = agree or strongly agree). To measure attitudes towards cancer screening (four questions in the positive domain and four in the negative domain), participants were asked to answer items on a five-point Likert-type scale, scored from 1 to 3 (for the positive domain: 1 = strongly disagree or disagree, 2 = neither agree nor disagree, 3 = agree or strongly agree; for the negative domain: 1 = strongly agree or agree, 2 = neither agree nor disagree, 3 = disagree or strongly disagree).

#### 2.2.4. Data Analysis

A descriptive analysis was conducted to report demographic, social, and health characteristics of participants. Chi-square tests were performed to evaluate differences among categorical variables for beliefs about cancer and cancer screening beliefs, and how these beliefs varied with CRC screening behaviour. A logistic regression model was used to identify independent factors that influenced CRC screening behaviour. Significance level was set at 0.05. The data analyses were performed with SPSS statistical software (version 21.0, IBM company, Armonk, NY, USA, 2014).

## 3. Results

### 3.1. Participant Characteristics

A total of 1104 unique surveys were submitted, of which 1048 met inclusion criteria, After excluding seven surveys with considerable missing data, the final sample included 1041 participants aged 35–74. We restricted our analyses to individuals aged 50 to 74 (*n* = 724). The characteristics of these participants are presented in Table 1. In this study, 77.46% [550/710] people were overweight or obese. About half of the participants rated their health as very good or excellent (48.62 [352/724]) and only one fifth reported that their life was quite a bit or extremely stressful (20.30% [147/724]). Half were from rural areas (51.24% [371/724]), just under one-third had ‘low’ education (high school or less; 27.92% [201/720]), and half had a household income of $60,000 or more (48.48% [318/656]).

Our sample was partially representative of the population, specifically of rural and urban geography, age, income, proportion of people who have a regular healthcare provider, and of people who have had a colonoscopy or sigmoidoscopy [12]. However, men and those with a lower education were underrepresented. We provide a more comprehensive analysis of representativeness elsewhere [12].

### 3.2. Associations between CRC Screening and Beliefs about Cancer Treatment and Outcomes

Most of the participants held positive, non-fatalistic attitudes towards cancer treatment and outcomes. There were no statistically significant differences between people who have ever had CRC screening and those who have never had CRC screening for any of the six beliefs about cancer treatment and outcomes studied (Table 2). While we did not control for sociodemographic variables, in additional analyses (not shown) we found only two beliefs had correlations with age or gender: older age was correlated with ‘believing that cancer can often be cured’ (r_s_ = 0.106, *p* = 0.005) and men were more likely than women to disagree with the belief ‘that cancer treatments are worse than the cancer itself’ (r_s_ = 0.097, *p* = 0.009). Interestingly, 61.3% (433/712) of people aged 50–74 agreed or strongly agreed that cancer treatment is worse than cancer itself, and 28.9% (209/720) agreed or strongly agreed that cancer is a death sentence. Almost all the participants (96.0% [692/721]) believed that going to see a doctor as quickly as possible after noticing a symptom of cancer could increase chances of survival.

### 3.3. Associations between CRC Screening and Beliefs about Cancer Screening

Table 3 displays comparisons in beliefs about cancer screening in individuals aged 50 to 74 to explore whether beliefs differed between people who engaged in CRC screening behaviours and those who did not. While Table 3 does not control for gender and age, there were no significant correlations with age among any of the cancer screening beliefs (analysis not shown), and there was only one screening belief (that they would participate in screening if their doctor told them how important it was) that was significantly correlated with gender (r_s_ = 0.082, *p* = 0.027), though the correlation was very small.

The vast majority of participants (84.5% [612/724]) agreed or strongly agreed that screening could reduce their chances of dying from cancer. Interestingly, differences in this attitude were not associated with differences in CRC screening behaviour (χ^2^(2) = 1.410, *p* = 0.501). Most participants (83.29% [598/718]) correctly disagreed or strongly disagreed with the statement that cancer screening was only necessary if they had symptoms. As shown in Table 3, those who had never had CRC screening were twice as likely to agree/strongly agree that they “would be so worried about what might be found during screening, that I would prefer not to do it” than those who have ever had CRC screening. The corresponding proportions are 9.2% and 4.9% (χ^2^(2) = 9.380, *p* = 0.009), respectively. The results suggest that fearing an undesirable screening outcome is a factor deterring people from receiving this screening service.

Furthermore, compared to those who have not had CRC screening, those who had CRC screening differed significantly in their agreement with the statement that screening was only necessary if one had symptoms (χ^2^(2) =15.680, *p* < 0.001). Those who never had screening were twice as likely to agree or strongly agree (12.5% vs. 5.4% for those who have had CRC screening) that screening was only necessary if they had symptoms (χ^2^(2) =15.680, *p* < 0.001). While 88.0% of people who have had screening disagreed or strongly disagreed with this statement, this was true for only 77.3% of those who have never had CRC screening.

Beliefs about whether screening has a high risk of leading to unnecessary surgery differed between people have had and those who have not had CRC cancer screening (χ^2^(2) =6.824, *p* = 0.032). Specifically, of people who had not had CRC screening, 11.9% agreed or strongly agreed that there was a high risk of unnecessary surgery, whereas only 7.1% of those who had screening agreed or strongly agreed. Overall, it appears that more of the ‘negative’ beliefs about screening were more associated with differences in screening behaviour than ‘positive beliefs’.

### 3.4. Logistic Regression of Associations between CRC Screening Behaviour and Sociodemographic Factors

Overall, 57.4% (409/713) of respondents aged 50 to 74 reported having ever had CRC screening. In the logistic regression model, we adjusted for gender, age, rural/urban geography, ethnicity, BMI class, whether they were living with a partner, their level of education, their income level, having had a history of cancer themselves, and having a first-degree relative with a history of cancer. Expectedly, there were differences in CRC screening by age group. Compared to individuals aged 50–54, those aged 55–59, 60–64, 65–69, and 70–74 all had significantly higher odds of having ever had CRC screening (Table 4). We also found that the odds of having had CRC screening were higher in males than females (OR = 1.689, 95% CI = 1.135–2.515). There were no statistically significant differences in CRC screening behaviour based on ethnicity, BMI class, geography, whether someone was living with a partner, education, income, or having a regular healthcare provider (Table 4). Furthermore, there were no differences based on whether the individual themselves was ever diagnosed with cancer, or whether they had a first-degree relative who had ever been diagnosed with cancer.

## 4. Discussion

Most participants held positive attitudes towards cancer treatment and outcomes. Positive attitudes towards, and awareness of, cancer and cancer screening play a role in adopting proactive strategies to prevent cancer, and so it is important for public awareness campaigns to emphasize the benefits of screening, while at the same time not catastrophising the disease or instilling fatalistic beliefs. Interestingly, none of the beliefs about cancer treatment or outcomes were associated with differences in CRC screening behaviour. This is encouraging because this suggests that even those who hold more fatalistic beliefs may still participate in screening.

Most participants also held positive beliefs about cancer screening, with few concerned about risks of false positives leading to unnecessary surgery, and even fewer who stated their fears about what might be found would prevent them from participating in screening. That said, we found those in our study who were worried about what might be found during screening were less likely to have ever had CRC screening, which supports previous research [9]. Interestingly, three of the four beliefs about cancer screening in the ‘negative domain’, but only one of the four beliefs in the ‘positive domain’, were associated with differences in behaviour. This suggests that awareness campaigns may benefit more from addressing fears or negative beliefs that participants have about screening, rather than just promoting the benefits. As those who never had CRC screening were more likely to believe screening was only necessary if they had symptoms, there may be some role for improving public awareness on the purpose of screening.

In our sample, only 55.9% of participants aged 50–74 agreed or strongly agreed with the statement that cancer screening is now very routine. However, since there was no association with having had CRC screening, this suggests that trying to change public perception on the routine nature of cancer screening may not improve screening participation. This is counterintuitive, but perhaps this finding is because other factors beyond beliefs are at play. Believing that regular screening would give them a feeling of control over their health was also associated with CRC screening. A large majority of participants agreed that they would participate in screening if their doctor told them how important it was, but this was not associated with a difference in CRC screening behaviour.

To find only a few small differences between screening beliefs and screening behaviours, and no differences in screening behaviours across the beliefs about cancer treatment and outcomes, should be considered a favourable finding. This is because it suggests that even NL residents who have fatalistic beliefs about cancer are barely, if at all, less likely to have participated in CRC screening. We interpret this as an encouraging sign that, despite their beliefs, people are still engaging in cancer screening. That said, Newfoundland and Labrador still has the second lowest rate, behind Quebec, of being up-to-date for CRC screening among all provinces in Canada [16]. Just over half of participants aged 50–74 have ever had FIT (or FOBT) or flexible sigmoidoscopy CRC screening (57.36% [409/713]). Due to the limitations of our non-random sampling design, this number should not be generalized as a population estimate. This also does not mean that these participants are up-to-date with screening. That said, these findings further support our assertion that there are gaps between attitudes and action. The lack of large associations between beliefs and CRC screening rates further suggests that interventions beyond health awareness and education may be necessary if public health campaigns wish to improve screening rates.

We identified few sociodemographic associations that were not statistically significant; however, previous research has identified that income and BMI class were associated with rates of being up-to-date on CRC screening [16,17]. That said, another study found little difference in ever-screening rates based on income or rural/urban geography [11]. In addition, among our participants, men unexpectedly had higher odds of ever having had CRC screening compared to women, whereas Singh et al. [16] found absolute rates of screening were slightly higher among women in NL, and that country-wide, the odds were no different. It is possible that the difference between our findings and that of the literature is due to the different outcome variables assessed, such that these factors may not play a role in having ever had CRC screening, but that they do play a role in being up-to-date with CRC screening. As expected, we found screening was lowest among individuals aged 50–54 and that, compared to this group, odds were more than double among people aged 55–59, and the odds increased with increasing age, with approximately triple odds among people aged 70–74.

There are a number of important limitations to note. We used Facebook advertising to recruit our sample, which is a non-random method and may thus lead to sampling bias, preventing generalizations to be made about prevalence. In particular, men were underrepresented in our sample. Using Facebook for recruitment undoubtedly led to a bias in that we sampled those who are technology literate enough to be using Facebook. There was also a selection bias, which is evidenced by the fact that more people were more health conscious (as defined by having an up-to-date flu vaccination status) and had higher levels of education [12]. This could have led to greater representation of those who are more likely to have had CRC screening. That said, even with random sampling methods, overrepresentation of more educated participants is common, and in our sample the proportion of participants who had ever had a colonoscopy or sigmoidoscopy was still the same as in the most recent CCHS survey [12]. Furthermore, our survey did not capture data on Indigenous identity, nor did it capture distribution of ethnic/racialized identities, and so these results are not generalizable to these populations and further study should attempt to characterise the barriers these already marginalised groups face. The cross-sectional nature of the study also limits us from making any conclusions about causation vs. correlation. Another limitation is that we looked at whether people had ever had screening, not whether they were up-to-date with screening. It is possible that the effect of beliefs and sociodemographic factors on being up-to-date with screening may be more or less significant than the effects on ever versus never having had screening. One additional consideration is that we assessed beliefs about cancer in general, but this paper explored CRC screening behaviour in particular. It is possible that beliefs and behaviours vary differently based on cancer type and that general beliefs about may differentially affect specific screening behaviours.

## 5. Conclusions

The majority of sampled residents in NL held positive beliefs towards cancer screening and non-fatalistic beliefs towards cancer treatment and outcomes. However, as a result of our non-randomised sampling method, and in particular the consequential over-representation of males, over-representation of higher levels of education, and lack of ethnic/racialized subgroup data, the generalisability of these findings is limited. Given the particularly high burden of CRC among the NL population, efforts to promote uptake of CRC programs should prove beneficial to population health. Further research into the relationship between beliefs and behaviour should attempt to confirm these findings and further elucidate this connection in an effort to optimise how public campaigns can promote cancer screening. Specifically, future studies should explore more complex causal mechanisms using structural equation modelling. Our findings suggest, however, that improving awareness and beliefs about cancer and cancer screening may be necessary but not sufficient to improve health behaviour. Previous research has suggested that behavioural health promotion often fails in reducing health inequities [18] and, as such, upstream action targeted at system-level changes may be more effective. Furthermore, given the considerable discordance between largely positive attitudes towards cancer screening and the relatively poor uptake of cancer screening among participants in our study, we stress that interventions targeted at beliefs and behaviours are necessary but not sufficient. Without system-level changes to reduce barriers to screening, we believe that behavioural interventions will likely yield less-than-favourable results.

## Figures and Tables

**Table 1 healthcare-10-02574-t001:** Characteristics of study participants aged 50–74.

Characteristics		Participants	
		*n*	%
Gender (n = 724)			
	Female	535	73.90
	Male	189	26.10
Age (n = 724)			
	50–54	157	21.69
	55–59	169	23.34
	60–64	179	24.72
	65–74	219	30.24
Ethnicity (n = 704)			
	Caucasian/White	677	96.16
	Other	27	3.84
Geography (n = 724)			
	Urban	353	48.76
	Rural	371	51.24
Living with a partner (n = 721)			
	No	167	23.16
	Yes	554	76.84
Education (n = 720)			
	High	519	72.08
	Low	201	27.92
Household Income (n = 656)			
	$60,000+	318	48.48
	$30,000–$59,999	225	34.30
	<$29,999	113	17.23
Health (n = 724)			
	Poor or Fair	120	16.57
	Good	252	34.81
	Very Good or Excellent	352	48.62
Stress (n = 724)			
	Extremely stressful	16	2.21
	Quite a bit stressful	131	18.09
	A bit stressful	307	42.40
	Not very stressful	219	30.25
	Not at all stressful	51	7.04
BMI (n = 710)			
	Underweight or Normal Weight	160	22.54
	Overweight	257	36.20
	Obese	293	41.27
Chronic Conditions (n = 724)			
	0	321	44.34
	1–2	293	40.47
	3+	110	15.19
Has a Regular Health Care Provider (n = 723)			
	No	59	8.16
	Yes	664	91.84
CRC Screening Behaviour (n = 713)			
	Has Ever Had	409	57.36
	Has Never Had	304	42.64
History of Cancer Diagnosis (n = 722)			
	No	579	80.19
	Yes	143	19.81
History of Cancer Diagnosis in First Degree Relative (n = 705)			
	No or Not Sure	153	21.70
	Yes	552	78.30

**Table 2 healthcare-10-02574-t002:** Associations Between CRC Screening *^a^* and Beliefs About Cancer Treatment and Outcomes Among Adults Aged 50–74 in Newfoundland and Labrador.

Beliefs about Cancer Treatment and Outcomes *^b^*	CRC Screening (n(%))				*χ*^2^(1)	*p*
	Never		Ever			
	Disagree	Agree	Disagree	Agree		
1. These days, many people with cancer can expect to live normal lives.	101 (33.2)	203 (66.8)	131 (32.0)	278 (68.0)	0.113	0.736
2. Cancer can often be cured.	44 (14.5)	260 (85.5)	59 (14.5)	348 (85.5)	<0.001	0.993
3. Going to the doctor as quickly as possible after noticing a symptom of cancer could increase chances of surviving.	12 (3.9)	292 (96.1)	17 (4.2)	390 (95.8)	0.023	0.878
4. Most cancer treatment is worse than the cancer itself.	111 (36.5)	193 (63.5)	168 (41.2)	240 (58.8)	1.590	0.207
5. I would not want to know if I have cancer.	267 (88.4)	35 (11.6)	375 (92.4)	31 (7.6)	3.203	0.074
6. A diagnosis of cancer is a death sentence.	223 (73.6)	80 (26.4)	283 (69.5)	124 (30.5)	1.401	0.237

*^a^* A person was considered to have ever had CRC screening if they have ever had either FIT/FOBT or a Flexible Sigmoidoscopy screening test. *^b^* Questions adapted from Simon, Forbes [13], reproduced with permission.

**Table 3 healthcare-10-02574-t003:** Associations Between CRC Screening *^a^* and Beliefs About Cancer Screening Among Adults Aged 50–74 in Newfoundland and Labrador.

Beliefs about Cancer Screening *^b^*	Had CRC Screening (*n* (%))						*χ*^2^(2)	*p*
	Never			Ever				
	Disagree	Neutral	Agree	Disagree	Neutral	Agree		
1. I would be so worried about what might be found during screening, that I would prefer not to do it.	238 (78.3)	38 (12.5)	28 (9.2)	355 (86.8)	34 (8.3)	20 (4.9)	9.380	0.009 **
2. Cancer screening is only necessary if I have symptoms.	235 (77.3)	31 (10.2)	38 (12.5)	360 (88.0)	27 (6.6)	22 (5.4)	15.680	<0.001 **
3. Cancer screening could reduce my chances of dying from cancer.	29 (9.5)	19 (6.3)	256 (84.2)	44 (10.8)	18 (4.4)	347 (84.8)	1.410	0.501
4. If I have a healthy lifestyle, I don’t need to worry about having regular cancer screening.	257 (84.5)	29 (9.5)	18 (5.9)	348 (85.3)	34 (8.3)	26 (6.4)	0.356	0.842
5. Cancer screenings are now very routine tests.	62 (20.5)	77 (25.4)	164 (54.1)	94 (23.0)	73 (17.9)	241 (59.1)	5.933	0.051
6. Cancer screening tests have a high risk of leading to unnecessary surgery.	189 (62.4)	78 (25.7)	36 (11.9)	288 (70.4)	92 (22.5)	29 (7.1)	6.824	0.032 *
7. Regular cancer screening would give me a feeling of control over my health.	27 (8.9)	57 (18.9)	218 (72.2)	21 (5.1)	48 (11.8)	339 (83.1)	12.255	0.002 **
8. I would be more likely to do screening if my doctor told me how important it was	35 (11.6)	57 (18.9)	209 (69.4)	42 (10.3)	57 (13.9)	310 (75.8)	3.955	0.140

*^a^* A person was considered to have ever had CRC screening if they have ever had either FIT/FOBT or a Flexible Sigmoidoscopy screening test. *^b^* Questions 1–3 were adapted from Simon, Forbes [13] and questions 4–8 were adapted from Beydoun, Khanal [15]. Reproduced with permission. *: *p* < 0.05, **: *p* < 0.01.

**Table 4 healthcare-10-02574-t004:** Logistic Regression of Associations Between CRC Screening Behaviour ^a^ and Sociodemographic Factors Among Adults Aged 50–74 in Newfoundland and Labrador (*n* = 596) ^b^.

Characteristics		OR (95% CI) ^c^
Gender		
	Female	1.00
	Male	**1.689 (1.135–2.515)**
Age		
	50–54	1.00
	55–59	**2.288 (1.405–3.726)**
	60–64	**2.399 (1.440–3.996)**
	65–69	**2.735 (1.591–4.699)**
	70–74	**3.087 (1.582–6.025)**
Ethnicity		
	Caucasian	1.00
	Other	1.230 (0.503–3.008)
BMI		
	<25.0 (Under or normal weight)	1.00
	25.0 to <30.0 (Overweight)	1.062 (0.673–1.678)
	30.0 + (Obese)	1.465 (0.940–2.285)
Geography		
	Urban	1.00
	Rural	1.158 (0.815–1.647)
Living with a partner		
	No	1.00
	Yes	1.013 (0.655–1.569)
Education		
	High (Post-secondary degree)	1.00
	Low (No post-secondary degree)	0.955 (0.639–1.428)
Income		
	$60,000+	1.00
	$30,000–$59,999	0.915 (0.607–1.379)
	<$29,999	0.932 (0.532–1.634)
History of cancer diagnosis in first-degree relative		
	No or not sure	1.00
	Yes	1.433 (0.947–2.169)
History of cancer diagnosis		
	No	1.00
	Yes	1.219 (0.782–1.898)

^a^ A person was considered to have ever had CRC screening if they had ever had either FIT/FOBT or a Flexible Sigmoidoscopy screening test. ^b^ The number of individuals included in the regression is smaller than the number of individuals aged 50–74 in our sample because some individuals had missing data and were thus not included in this analysis. ^c^ Bold text indicates statistical significance at the level of 0.05 after adjusting for gender, age, geography, ethnicity, BMI class, living with partner, education category, income category, history of cancer in self, and history of cancer diagnosis in first degree relative.

## Data Availability

The data generated during and/or analyzed during the current study are available from the corresponding author on reasonable request.

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
