# Peer review of "The Effects of Cancer Beliefs and Sociodemographic Factors on Colorectal Cancer Screening Behaviours in Newfoundland and Labrador"

_healthcare, 2022, doi:10.3390/healthcare10122574_

Round 1

Reviewer 1 Report

This is an interesting study and a deserving topic, and is generally well written.

The sample used is problematic - recruiting subjects via Facebook, particularly when targeting an older demographic runs the risk of selecting subjects who are 1) more technologically literate than average and 2) more aware of CRC risk at baseline and thus more likely to have had CRC screening.  Given this confounding factor, more information regarding the sample and it's representation of the NL population is needed, rather than referring the reader to another paper.

The introduction should contain some comparison of the CRC rate and screening rate in NL v.s. other provinces and/or the country overall.

Can Table 2 be presented in a more reader-friendly way such as a graph to illustrate the distribution of responses between screened and unscreened subjects?

Conclusions re gap between attitudes and action are potentially important and point to other barriers to screening in this population, BUT need to sure that these conclusions are drawn from a representative sample of the population.

Author Response

Q1:  The sample used is problematic - recruiting subjects via Facebook, particularly when targeting an older demographic runs the risk of selecting subjects who are 1) more technologically literate than average and 2) more aware of CRC risk at baseline and thus more likely to have had CRC screening.  Given this confounding factor, more information regarding the sample and it's representation of the NL population is needed, rather than referring the reader to another paper.

A1: We have included a few sentences in the methods section on page 3 to better describe the sample and in which ways it was and was not representative of the population.

“The sample was over-representative of women, those with post-secondary education, and those who had an up-to-date flu shot. It was representative rural-urban geography, those with a regular healthcare provider, and those who have had a colonoscopy[12]. The sample was practically representative of age distribution, household income, smoking status, and BMI[12]. “

 We have also included additional comments in the limitations section of our paper describing the selection bias that you mention. We do note, however, that the selection bias is also inherent in any sampling method and it is common for those with more education to be overrepresented. We hope that mentioning these limitations helps satisfy the reviewer’s concerns.

“In particular, men were underrepresented in our sample. Using Facebook for recruitment undoubtedly lead to a bias in sampling those who are technology literate enough to be using Facebook. There is also a selection bias, which is evidenced by the fact that more people were more health conscious (as defined by having an up-to-date flu vaccination status) and had higher levels of education[12]. This could have lead to greater representation of those who are more likely to have CRC screening. That said, even with random sampling methods, overrepresentation of more educated participants is common, and in our sample the proportion of participants who ever had a colonoscopy or sigmoidoscopy was still the same as in the most recent CCHS survey[12]. Furthermore, our survey did not capture data on Indigenous identity, nor did it capture distribution of ethnic/racialized identities, and so these results are not generalizable to these populations and further study should attempt to characterize the barriers these already marginalized groups face.” 

Q2:  The introduction should contain some comparison of the CRC rate and screening rate in NL v.s. other provinces and/or the country overall.

A2: The introduction already includes the following statement

The uptake of screening has been far less than desirable in Canada, and it is particularly poor in regions of Newfoundland and Labrador[11].

We have added an additional statement with further elaboration of those rates provided by the study referenced:

“Across 43 health regions in Canada, never-use rates of FOBT ranged from 53.3 to 89.2%, and never-use of endoscopy ranged from 81.1 to 94.3%, whereas the never-use rates in NL health authorities ranged from 83.6 to 89.1% for FOBT, and 84.6 to 94.3% for endoscopy[11], though this was before introduction of the new FIT screening kits.”

Q3: Conclusions re gap between attitudes and action are potentially important and point to other barriers to screening in this population, BUT need to sure that these conclusions are drawn from a representative sample of the population.

A3: As mentioned above and in the paper, our sample has limitations given it was a non-random sampling method, and it is representative of some key sociodemographic and health characteristics of the population, but it is not representative of others. Even among random sampling methods, it is common to have a selection bias leading to greater numbers of women, those with higher incomes, and those with higher levels of education and, consequently, health awareness, be over-represented [1-4]

  1. Korkeila K, Suominen S, Ahvenainen J, Ojanlatva A, Rautava P, Helenius H, et al. Non-response and related factors in a nation-wide health survey. Eur J Epidemiol 2001;17(11):991-999. [CrossRef] [Medline]
  2. Etter JF, Perneger TV. Analysis of non-response bias in a mailed health survey. J Clin Epidemiol 1997 Oct;50(10):1123-1128. [CrossRef] [Medline]
  3. Etter JF, Perneger TV. Analysis of non-response bias in a mailed health survey. J Clin Epidemiol 1997 Oct;50(10):1123-1128. [CrossRef] [Medline]
  4. Richiardi L, Boffetta P, Merletti F. Analysis of nonresponse bias in a population-based case-control study on lung cancer. J Clin Epidemiol 2002 Oct;55(10):1033-1040. [CrossRef] [Medline]

The limitations of this are explained in our paper and we do not believe they have significant bearing on our conclusion that system-level changes to reduce barriers to screening are needed. It is not possible to change the sampling method at this stage, and our initial decision to use this non-random method was based on financial and time limitations. We hope that by explaining these limitations in our study, and framing the conclusions we draw in that context, we can still offer the literature these valuable findings.

Q4: Can Table 2 be presented in a more reader-friendly way such as a graph to illustrate the distribution of responses between screened and unscreened subjects?

A4: Tables have been formatted a little better.

Reviewer 2 Report

I find the results encouraging, which guide how to inform the population and which should be published. I have some comments to clarify some parts of the text.

Author Response

Q1. “Why are these questions considered relevant? Please explain the initial reasoning and the hypothesis raised to carry out the questionnaire with these questions, and do not include questions related to diet or lifestyle, which possibly greatly influence the onset of the neoplastic event?”

The questions related to participant information were focused on sociodemographic factors because of their significant importance to health outcomes, and because we wished to understand if these were also determinants of health behaviours. Numerous studies completed by our PI in the past have looked at factors related to diet and lifestyle [e.g., 4-5], but these are only partially responsible for CRC burden and, I’m sure you would agree, globally sociodemographic factors continue to the greatest burden of disease. We wanted to focus our study on more of the upstream determinants of health. This was driven by our third research objective: “(3) Analyze independent associations between sociodemographic factors and CRC screening behaviour”. Our hypothesis was that those with more resources would be more likely to engage in CRC screening, as they would be more financially stable and thus more able to turn their attention away from the day to day tasks of ‘surviving’/making enough money to put dinner on the table, to put it plainly. Alas, we stated in our discussion on Page 9 that this was an unexpected finding “That we identified few sociodemographic associations was unexpected, however, as previous research has identified that income and BMI class were associated with rates of being up-to-date on CRC screening[16, 17].”

  1. Chen, Z., Wang, P. P., Woodrow, J., Zhu, Y., Roebothan, B., Mclaughlin, J. R., & Parfrey, P. S. (2015). Dietary patterns and colorectal cancer: results from a Canadian population-based study. Nutrition Journal, 14(1), 8. https://doi.org/10.1186/1475-2891-14-8
  2. Zhao, J., Zhu, Y., Wang, P. P., West, R., Buehler, S., Sun, Z., Squires, J., Roebothan, B., McLaughlin, J. R., Campbell, P. T., & Parfrey, P. S. (2012). Interaction between alcohol drinking and obesity in relation to colorectal cancer risk: a case-control study in Newfoundland and Labrador, Canada. BMC Public Health, 12, 94. https://doi.org/10.1186/1471-2458-12-94

Q2. Please include in complementary data the surveys carried out and their evaluation.

As in our data availability statement, the data generated during and/or analyzed during the current study are available from the corresponding author on reasonable request. We would be happy to include our survey instrument as a supplementary file and have included it now. Under section 2.2.1. Participant Information, we have now added the following sentence.

“See Supplementary File 1 for our survey instrument.”

Q3. Do you consider that the presence of more women in the study could be an statical bias?

Indeed, it likely is a statistical bias and we have included an additional paragraph in our discussion to explore these limitations, and related sampling biases.

“In particular, men were underrepresented in our sample. Using Facebook for recruitment undoubtedly lead to a bias in sampling those who are technology literate enough to be using Facebook. There is also a selection bias, which is evidenced by the fact that more people were more health conscious (as defined by having an up-to-date flu vaccination status) and had higher levels of education[12]. This could have lead to greater representation of those who are more likely to have CRC screening. That said, even with random sampling methods, overrepresentation of more educated participants is common, and in our sample the proportion of participants who ever had a colonoscopy or sigmoidoscopy was still the same as in the most recent CCHS survey[12]. Furthermore, our survey did not capture data on Indigenous identity, nor did it capture distribution of ethnic/racialized identities, and so these results are not generalizable to these populations and further study should attempt to characterize the barriers these already marginalized groups face.”

We also have decided to further explicitly state the limitations in our conclusion, adding the word “sampled” to the first sentence, and then another sentence to our opening of our Conclusion section:

“The majority of sampled residents in NL held positive beliefs towards cancer screening and non-fatalistic beliefs towards cancer treatment and outcomes. However, as a result of our non-randomized sampling method, and in particular the consequent over-representation of males, over-representation of higher levels of education, and lack of ethnic/racialized subgroup data, the generalizability of these findings is limited.”

Q4. When the authors propose this conclusion: "The results suggest that fear of an undesirable screening result is a factor that discourages people from receiving this screening service", I would ask that they justify it well, because if this is indeed what happens, it would be It is necessary that measures be proposed in the article to avoid this disastrous procedure. Since patients are asked for information on how informed they are about the consequences, this result should be related to patients who are or are not well informed about the consequences of early and late tumor detection.

We are not entirely sure what point the Reviewer is trying to communicate here. Our study directly asked patients whether they agree or disagree with the statement "would be so worried about what might be found during screening, that I would prefer not to do it”. Many more patients who never had CRC screening agreed with that statement than disagreed with it. Generally speaking, if a patient is worried about the results of the test, worry and fear are pretty synonymous, and so they could be said to ‘fear the results of the test’. There was an obvious association between fearing results of the test and not having ever had the test done. We do not believe this requires much more explanation than what we have afforded it in the manuscript, as this is a fairly simple interpretation of the data. Fearing the outcome of screening as a reason not to engage in cancer screening has been described in the literature previously, and in our discussion we do discuss this and cite a study that supports this. For the Reviewer here, we have added an additional sentence in our conclusion to tie this in and have referenced a supporting study:

“That said, we found those in our study who were worried about what might be found during screening were less likely to have ever had CRC screening, which supports previous research [9].”

Q5. Concisely add the conclusion that the authors draw from this part of the study (Section 3.3)

We have added the following sentence in Section 3.3 :

“Overall, it appears that more of the ‘negative’ beliefs about screening were more associated with differences in screening behaviour than ‘positive beliefs’.”

This is expanded upon more in the discussion, which already states: “Interestingly, three of the four beliefs about cancer screening in the ‘negative domain’, but only one of the four beliefs in the ‘positive domain’, were associated with differences in behaviour. This suggests that awareness campaigns may benefit more from addressing fears or negative beliefs participants have about screening, rather than just promoting the benefits.”

Thank you for recommending we also summarize the subsection of the results. We believe this much better ties in our results to our discussion section, and the subsequent conclusions/recommendations we made.

Q6. Add a comment if the authors expected to have differences with demographic factors or not and why? (to Results section 3.4)

Thank you for your input here. However, we believe that since this is the results section, it would not be the appropriate place to discuss our expected results. We do, however, discuss them in our Discussion section, in the paragraph that starts with:

“That we identified few sociodemographic associations was unexpected, however, as previous research has identified that income and BMI class were associated with rates of being up-to-date on CRC screening[16, 17].” …

In that paragraph, we further expand on our findings, how they compare to the literature, and the reasons we believe why we did not find many differences with demographic features in our study. 

Q7. I find the results encouraging, which guide how to inform the population and which should be published. I have some comments to clarify some parts of the text. (You left this comment on Top of Page 9)

Thank you. We appreciate your support in reviewing this article. We have addressed the other comments you made, and hope they satisfy you.

Q8. How important the authors are to this limitation within the study: "Another limitation is that we looked at whether people had ever been screened, not whether they were up-to-date with screening. It is possible that the effect of beliefs and sociodemographic factors of being up to date with screening may be more or less significant than the effects of ever versus never having been screened." (this comment was in the limitations section of our discussion)

To assess whether patients were up to date with screening would have been another valuable thing to look at, but for the purposes of our study we decided to simplify the focus to those who had or had never participated in screening. We suspected that a dichotomous outcome of ever vs never had screening would be more likely observed if it were based on beliefs about screening in general, where we expected there to be many more factors at play in determining whether someone was up to date on cancer screening (e.g. experiences with the last screening test, whether their primary care provider was proactive in reminding them, etc.), and so we were interested mostly in those who had never gotten screening before, and if it were their beliefs about cancer screening which may have contributed. As our focus of the study was stated as looking at ever vs never having had CRC screening, and this limitation is mentioned, we do not feel there is a need to explain it further in the limitations section and hope the reviewer agrees with this. 

Reviewer 3 Report

General

The authors present a well conducted and interesting study dealing with an important clinical and scientific problem. New data are well and transparently presented.

However, there are remaining questions:

I wonder why the authors did not deal with the differences between non-invasive techniques of CRC screening (FOBT, FIT) and invasive ones (sigmoidoscopy), because obviously the have the respective data. The barrier between the two mentioned methods of screening differs and it would be of interest whether the asked beliefs are correlated with the two screening methods.

Furthermore I wonder about the low percentage of participants with “other ethnicity”? Is the indigenous population in these districts of Canada correctly represented?

Specific

Line 158 : What means “unexpectedly”? Significantly?

Line 169: The male underrepresentation seems to be enormous. How can this problem be smoothened by the cited publication (12)?

Author Response

Q1: I wonder why the authors did not deal with the differences between non-invasive techniques of CRC screening (FOBT, FIT) and invasive ones (sigmoidoscopy), because obviously the have the respective data. The barrier between the two mentioned methods of screening differs and it would be of interest whether the asked beliefs are correlated with the two screening methods.

A1: Most important to us was whether to investigate the associations between our independent variables and CRC screening behaviour in general. There are undoubtedly barriers to either method, but we felt these were not the barriers we believed were as important to look at. Furthermore, in some patients it is not clinically appropriate to do FIT testing (e.g. patients with Inflammatory Bowel Disease, patients with a previous positive endoscopy who needs a surveillance follow-up), so it may mean some people are not technically choosing one method over another but has some reason that requires them to use one method. Most of all, if someone is engaging in CRC screening, so long as they are using a clinically appropriate method, for this study we cared less if it was invasive or non-invasive. We wanted to look at the barriers to engaging in any screening method, period. It might be worth examining in a future analysis, though, so we thank the reviewer for recommending they would find it worthy of interest.

Q2:Furthermore I wonder about the low percentage of participants with “other ethnicity”? Is the indigenous population in these districts of Canada correctly represented?

A2: Unfortunately, this study was not representative of racialized/ethnic populations, and we did not provide sufficient options to capture this data at the time our survey was designed, which is an oversight on our part. The Indigenous population in NL is 11.4% based on the 2016 census. The prevalence of visible minorities (which does not include Indigenous populations, by Statistics Canada definitions) in NL is 2.3%. We have added an additional limitation on page 10 clarifying that these results are not generalizable to these populations:

“Furthermore, our survey did not capture data on Indigenous identity, nor did it capture distribution of ethnic/racialized identities, and so these results are not generalizable to these populations and further study should attempt to characterize the barriers these already marginalized groups face.”

Q3:  Line 169: The male underrepresentation seems to be enormous. How can this problem be smoothened by the cited publication (12)?

A3: It is not uncommon for men to have lower rates of participation in health research, even randomly sampled health research, but we agree that our sample is especially underrepresented. Unfortunately, we do not see anything that can be done beyond explicitly mentioning this. We have clarified our conclusion by adding “sampled” in the first sentence of the conclusion:

“The majority of sampled residents in NL held positive beliefs towards cancer screening and non-fatalistic beliefs towards cancer treatment and outcomes.”

We have also added the following sentence immediately after the above sentence into the conclusion:

“However, as a result of our non-randomized sampling method, and in particular the consequent over-representation of males, over-representation of higher levels of education, and lack of ethnic/racialized subgroup data, the generalizability of these findings is limited.”

In retrospect, we should have extended the study period to spend more time running targeted ads at men to better recruit this population, but we hope that clarifying these limitations very explicitly will help to smoothen and better contextualize our findings. Despite these limitations, we believe they still have validity to the sampled population and those members of the population that our sample better represents, and that they are important findings.

Q4:Line 158 : What means “unexpectedly”? Significantly?

A4: The sentence that gave rise to ambiguity has been removed.

Round 2

Reviewer 3 Report

The authors followed the suggestions of the reviewers carefully and improved the presentation significantly. Thank you !!